# Echogenic intracardiac foci detection and location in the second-trimester ultrasound and association with fetal outcomes: A systematic literature review

**Hope Eleri Jones** [1] *, **Serica Battaglia**[2], **Lisa Hurt**[3], **Orhan Uzun**[4], **Sinead Brophy** [1]

1 National Centre for Population Health and Wellbeing Research, Swansea, United Kingdom, 2 Swansea University, Sketty, Swansea, United Kingdom, 3 Cardiff University, University Hospital of Wales, Cardiff, United Kingdom, 4 Cardiff and Vale UHB: NHS Wales Cardiff and Vale University Health Board, Children's Heart Unit, Cardiff, United Kingdom

* h.e.jones@swansea.ac.uk

**Data Availability Statement:** This is a literature review so the relevant data is in each of the papers mentioned in the review that are all open access.

## Abstract

### Background

Echogenic Intracardiac Foci (EIF) are non-structural markers identified during the routine 18–20-week foetal anomaly ultrasound scan yet their clinical significance on future outcomes for the infant is unclear.

### Objective

To examine the association between EIF and risk of preterm birth, chromosomal abnormalities, and cardiac abnormalities.

### Design

A review across four databases to identify English language journal articles of EIF using a cohort study design. All studies were reviewed for quality using the Critical Appraisal Skills Programme (CASP) checklist and data extracted for comparison and analysis.

### Results

19 papers from 9 different countries were included. Combining these studies showed 4.6% (95% CI = 4.55–4.65%) of all pregnancies had EIF which was on the left in 86% of cases, on the right in 3% of cases and bilaterally in 10%. There was no evidence that EIF was associated with higher rates of preterm birth. However, it is possible that infants with EIF were more likely to be terminated rather than be born preterm as there was a 2.1% (range 0.3–4.2%) rate of termination or death of the foetus after week 20 among those with EIF. There was no evidence that EIF alone is highly predictive of chromosomal abnormalities. There was evidence that EIF is associated with higher rates of minor cardiac abnormalities (e.g. ventricular septal defect, tricuspid regurgitation or mitral regurgitation)) with 5.1% (224 of 4385) of those with EIF showing cardiac abnormalities (3.08% in retrospective studies and

The values behind the measures reported and used to build graphs and figures are provided in the supplementary information.

**Funding:** The author(s) received no specific funding for this work.

**Competing interests:** The authors have declared that no competing interests exist.

17.85% in prospective studies). However, the risk of cardiac defects was only higher with right-sided EIF and where the EIF persisted into the third trimester. However, this is a rare event and would be seen in an estimated 4 per 10,000 pregnancies.

## Conclusion

EIF alone was not associated with adverse outcomes for the infant. Only persistent EIF on the right side showed evidence of carrying a higher risk of cardiac abnormality and would warrant further follow-up.

## Introduction

Congenital heart disease (CHD) is characterised by defects in the heart as it forms in the growing foetus [1]. CHD is the most common congenital abnormality in infants today with a diagnosis made 1 in every 100 births [2, 3]. CHD is the leading cause of infant mortality [3, 4]. Improvements in early detection and diagnosis would greatly improve the support and service provision for families when the infant is born, developing early preventative interventions, and putting in place enhanced monitoring and care for the infant.

A foetal anomaly scan is offered to all pregnant women in the UK between weeks 18-to-20 of pregnancy which includes routine screening for CHD [5, 6]. However, it is estimated that only 50% - 75% of cases of serious congenital cardiac abnormalities are detected during this scan [5]. Potential discrepancies may be due to variations in equipment and facilities available to undertake the appropriate level of screening [7]. The extensive use and improvements to ultrasound imaging has enabled more detailed detection of fetal structural variations alongside the identification of more subtle non-structural findings, labelled as markers [7–9]. This has led to the identification of ultrasound soft markers (USM) [10], including Echogenic Intracardiac Foci (EIF), which are often discovered as an incidental finding during prenatal screening [11].

EIF have been described as small echogenic areas of microcalcification or fibrosis on the papillary muscle of either or both atrioventricular valves of the foetal heart [12]. EIF are often referred to as echo bright spots or hyper echogenicity of foetal soft tissue [13, 14] when visualised on a foetal ultrasound. They are of equal or greater brightness than the surrounding bone [15, 16]. The pathological cause of EIF remains ambiguous and it is still unclear what is the significance, if any, in terms of defects in the developing heart of the foetus. Research has suggested that EIF prevalence during this second-trimester scan is estimated to occur in between 0.17% and 20% of pregnancies [7]. The prevalence varies by race, with the highest incidence found amongst Asian populations [17].

Previous research has described EIF as often transient, resolving by the third trimester of pregnancy or after delivery [18], nonpathological or benign in isolation [11], and therefore has been highlighted that isolated soft markers may be present in 10% of normal foetuses [19]. However, multiple USM and the presence of specific soft markers, in terms of the specific location and size, may be associated with underlying chromosomal abnormalities (such as an association between the presence of EIF and a later diagnosis of Down's syndrome) albeit the current evidence is not supportive of this [20] or adverse infant outcomes [21]. This suggests that the presence of specific soft markers may be associated with structural abnormalities and developmental conditions [22].

Given that EIF are small structures within the foetal ventricle chamber, they have been commonly associated with other soft markers [19]. However, it is unclear whether EIF can be used as a screening tool for cardiac heart diseases as previous research has shown mixed results. Despite the possible concerns surrounding the association between the development of structural and chromosomal abnormalities in the affected foetus, available data is debatable and inconclusive in some publications whilst others clearly showed no significance.

The presence of EIF with additional risk factors such as older maternal age (>35 years) places the mother in a high-risk category [22] during the pregnancy. Depending on the association with other soft markers this may lead to a referral for an echocardiogram and/or genetic counselling for amniocentesis to investigate possible chromosomal abnormalities. If EIF persists until birth and is present alongside other soft markers, a postnatal echocardiogram would be recommended and a referral to neonatal consultant. However, with increasing maternal age [23] and improvements in ultrasound detection [9] this is becoming more common therefore adding to costs of healthcare, potentially with little gain as the prevalence of congenital abnormality has remained constant [24].

It remains unclear whether EIF encompasses both benign transient structural variations and pathological changes [18, 22]. Distinguishing between these two would be significant in terms of determining those to take forward for echocardiography and this is especially important in times of financial strain on the NHS. It is unclear whether location of EIF and number of EIFs are of more significance rather than their presence or absence.

The aim of this literature review is to examine the association between EIF and infant outcomes. Specifically: 1) The association between EIF identified at 18–20 weeks and risk of preterm birth, death, specific cardiac diagnosis (CHD) and chromosomal abnormalities. 2) The association between the number of EIF (single vs multiple), the location of EIF (left, right or bilateral) and the risk of preterm birth, death, cardiac diagnosis, and chromosomal abnormalities.

## Methods

This review follows the Population-Interest-Comparator-Outcome (PICO) criteria to guide the scope and breadth of the literature review and set the inclusion criteria. The population was defined as the foetus at the 18–20-week ultrasound scan. The interest was where EIF was identified in the scan (as identified by the sonographer) compared to the comparator of those with no EIF. The infant outcomes include preterm delivery, death, structural cardiac abnormality, and chromosomal abnormalities.

### Inclusion criteria

Peer-reviewed cohort studies in English, with full-text availability, published between 2013–2023. Studies included prospective and retrospective cohort studies with the assessment of EIF (with or without other markers) recorded by ultrasound at 18–20 weeks gestation and recording through follow-up (e.g., birth and post birth up to 5 years) with one of the outcome variables of interest. Studies which contain follow-up data to birth and infancy following assessment of routine ultrasound screening in a healthcare setting in the second trimester of pregnancy were included. Studies that only followed up EIF pregnancies without a comparison group were included in selected studies. However, if selection bias was present (e.g., only women >35 years) this was recorded in the data quality assessment and incorporated into the interpretation.

## Exclusion criteria

Studies were excluded if the full text was not available in English, they examined non-cardiac echogenic focus (e.g. identified within the kidneys, liver, bowel, thyroid, hepatic, and pancreas). Also excluded were those not performed during the second-trimester ultrasound scan or those studies predicting echogenic foci as the outcome rather than the exposure.

## Outcome measures

The primary outcome measure is preterm birth (before 37 weeks) or death of the foetus. Secondary outcomes include chromosomal abnormality (genetic testing for aneuploidy, amniocentesis or NIPT (Non-Invasive Prenatal testing) reports) and cardiac abnormality (fetal echocardiology reports and/or postnatal diagnosis).

## Literature search

The literature search was carried out in three stages: Stage one was an initial search on Pubmed to identify the index terms which best identify relevant articles. This initial search used the first 10 relevant primary studies and collated the search terms that were used. The second stage took these updated search terms to apply to the; Web of Science online tool which searches the following databases: Web of Science Core Collection, BIOSIS Citation Index, Current Contents Connects, Data Citation Index, Derwent Innovations Index, and MEDLINE (Medical Literature Analysis and Retrieval System Online). This search was repeated on Pubmed and Scopus individually. The final stage reviewed the reference list of the included studies and reviews in the area. Grey literature was also reviewed including the National Congenital Heart Disease Audit (NCHDA). The literature search was carried out in June 2023 and imported onto Covidence software [25]. The keywords identified for the initial search included: Echogenic intracardiac foci, Chromosomal abnormalities, Prenatal diagnosis, Ultrasonography, Echocardiography, Pregnancy trimester second, Foetal cardiac anomalies OR foetal cardiac abnormalities, Foetal echocardiography, Foetal heart, Trisomy 21 (Down syndrome), Prenatal screening and Pregnancy outcomes. The search terms used to identify relevant papers consisted of four concepts and were combined to form the following search string across all key databases: "echogenic intracardiac foc*" OR "echogenic foc*" OR "intracardiac foc*" OR "cardiac echogenic foc*".

## Study selection

Covidence automatically removes duplicate records [25]. Initial screening based on the title and abstract was carried out by two independent reviewers based on the inclusion and exclusion criteria. For those publications selected to take forward, a second screening based on the full text was conducted and those selected as relevant were taken forward to data extraction (Fig 1).

## Extraction of relevant variables

A data extraction form was designed which included the Critical Appraisal Skills Programme (CASP) quality review [26] to assess risk of bias. Quality was assessed to include methods of addressing confounding variables, quality of measurement of outcome and exposure and if outcomes by exposure and loss to follow-up. The data includes study identification (number, first author), study demographic information (country, size of study, study design), total number of pregnancies, number with EIF recorded, duration of follow up, recording of outcomes of interest, position (left/right) of EIF and number of EIF (single/multiple). See the

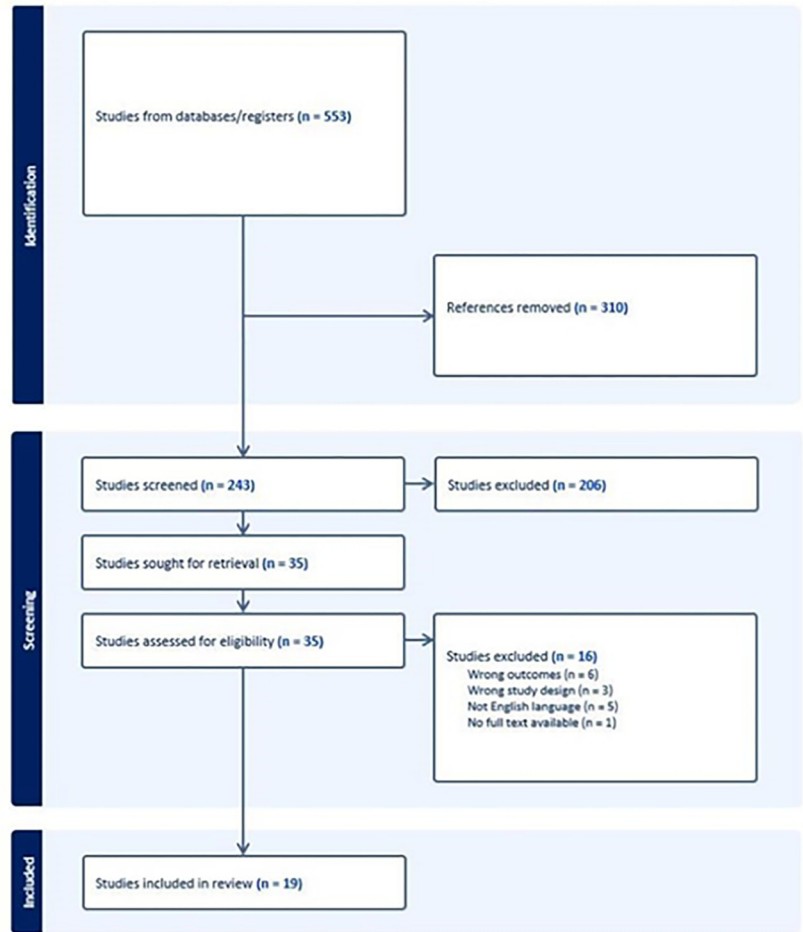

**Fig 1. PRISMA flow diagram.**

supplementary information for details of extracted variables, definitions and how they were recorded (S1 Table).

## Comparison data

The outcomes for those with no EIF were recorded in the same way as those with EIF reporting the number of preterm births, deaths, congenital abnormalities and cardiac abnormalities.

## Results

There were 243 non-duplicate studies that were identified using the search criteria. Of these, 206 were excluded at the title and abstract screening due to studies being published before 2013, not a study examining EIF (e.g., assessing cysts or other soft markers) and non-English studies. 35 full journal articles were reviewed. However, the majority were excluded on the basis that they did not have a cohort design (health economics study or case study or case series), did not have follow-up of those with EIF, or did not have follow up of the outcomes which were considered within this review. After full text review, 19 studies were included (Fig 2). These 19 studies covered 9 different countries with 9 of the studies from China, 2 from Iran and 2 from Pakistan. The smallest study had a follow-up of 7 EIF pregnancies and the largest

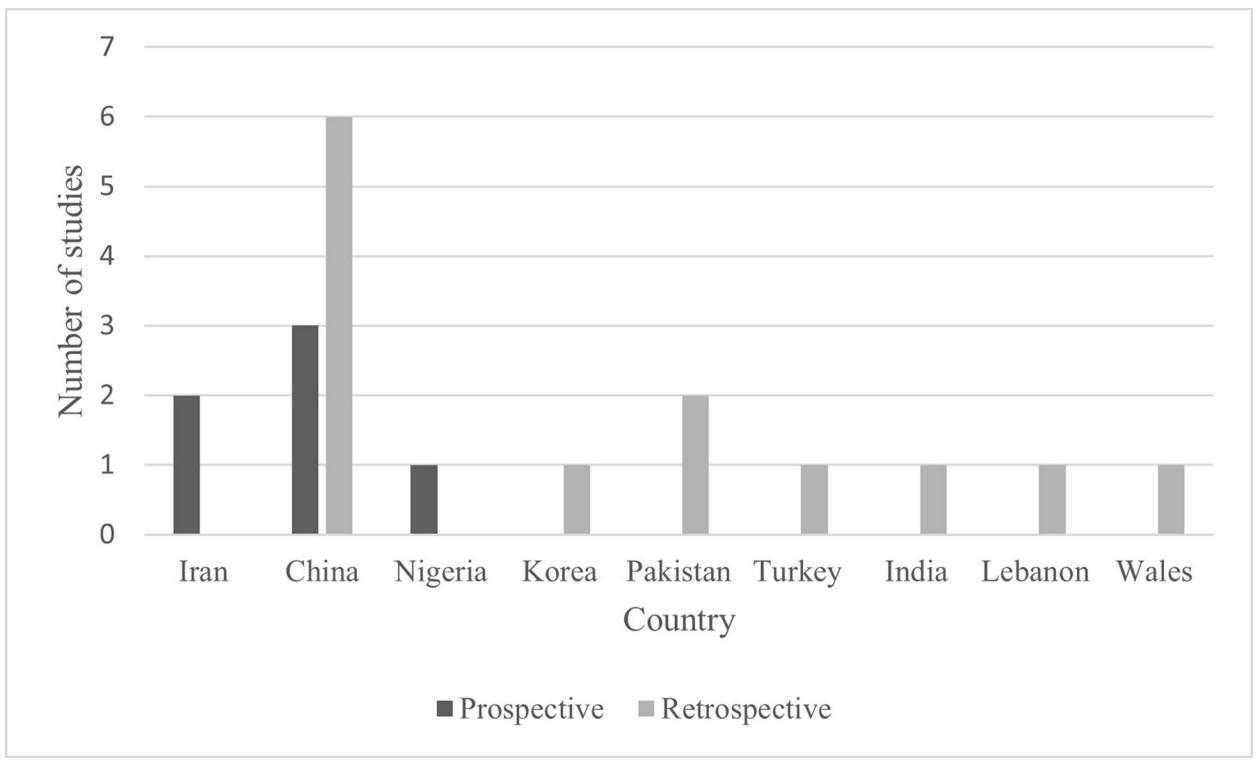

**Fig 2. Included studies.**

was 143,067 pregnancies of which 2647 had EIF. 6 studies had prospective follow-up of infants and the remaining 13 had retrospective follow-up using routine medical records.

The majority of studies were rated as being of moderate quality as there was missing data, or loss to follow-up, especially in the prospective studies. For example, a postpartum follow-up study was performed on women referred for echocardiography and of these 7 had EIF [1]. This study was rated as high risk of selection bias and so rated moderate. Another study was rated moderate as there was loss to follow-up among some EIF participants [27]. Other studies were not stratified by EIF characteristics making interpretation difficult and therefore rated low [28]. Moreover, in some cases the dates of recruitment were not being reported, and it was unclear how many terminations of pregnancies occurred [29]. Overall, the study quality was rated as moderate and there was a risk of bias such as loss to follow-up or selection bias in various papers.

### EIF association with outcomes

This review followed 34,104 women from 19 studies (4 studies did not provide a total number of women screened). Of these, there were 15,600 EIFs or 4.6% (95% CI = 4.55–4.65%) of all pregnancies and 4.5% (range 0.5% to 17.8%) of pregnancies when the EIF proportion of each study was averaged. This highlights that EIF is a relatively common finding on ultrasound scans. In addition, 5 studies followed-up those with EIF only (e.g. a cohort of those with EIF) which identified an additional 1,400 EIFs for follow-up. On average, 26.6% presented with multiple EIFs and the remaining had a single EIF. Taking the averaged levels in each study (6 studies reporting the location of EIF), the EIF was on the left side in 86% of cases, on the right in 3%, and bilaterally in 10%. Therefore, EIF on the right is a rare occurrence that only occurred in 13% of EIFs or 6 per 1000 pregnancies.

## Preterm births

Five studies reported the rates of preterm birth where the prevalence was 7.2% (range 2–13%). This rate is comparable to the UK where the preterm rate is 7.8% (23) compared to 13% in North Africa [30] In two studies that reported preterm rates among non-EIF pregnancies, the range was between 5–20%. Comparing the two papers that reported EIF and non-EIF preterm rates, there was no evidence that EIF is associated with preterm birth with 4.8% (EIF) versus 5.1% (non-EIF) in Wales with the sample size 18,841 EIF (N = 858) [31], and 13.0% (EIF) versus 20.9% (non-EIF) [32] with the sample size 9,270 (EIF = 230) in Lebanon. This follow-up in a total of 5 studies (2 with control data) does not provide evidence of higher preterm rates in those with EIF. However, in the study in the UK [31], the authors had the scans reviewed by a quality assurance panel and of the 858 EIF scans only 615 were assessed by the quality assurance panel as EIF's. This means that there were 44 preterm births out of 615 with EIF (7.2%) which is in line with population level preterm rates in Wales [23]. The non EIF was 921 preterm births out of 18,226 pregnancies without EIF (5.1%).

However, it is possible that infants with EIF were more likely to be foetal deaths or termination rather than being born preterm. There was a 2.0% (range 0.3–4.2%) rate of termination or death of the foetus after week 20, among those with EIF. This rate is higher than previously reported [33] which suggests that less than 0.5% of pregnancies will end in foetal death after 20 weeks. The main cause of death in China and Iran appears to be termination due to suggested chromosomal abnormalities and other soft markers. For example, pregnancies were terminated due to abnormal chromosome numbers (35)(28) and 50% of deaths had chromosomal abnormalities [19]. Many pregnant women with EIF had invasive foetal testing to screen for chromosomal abnormalities [28]. The findings presented that the increase in the risk of chromosomal abnormalities in the foetus was related to the pregnant women belonging to high-risk groups, rather than EIF [28].

In Wales, foetal deaths (0.48% of confirmed EIF pregnancies) were due to stillbirths and induced or spontaneous pregnancy loss. This rate of foetal death is in line with expected rates and comparable to non-EIF rates. In Lebanon, foetal deaths (0.9%) were due to foetal demise but this was at the expected rate and equivalent to non-EIF foetal loss rates. In Pakistan, the deaths were associated with cases of pulmonary artery hypertension and tetralogy of Fallot and were all neonatal deaths (Fig 3). This was a higher-than-expected rate, but the sample size was small.

Overall, there was no evidence that preterm rates were higher among those with EIF. Terminations were higher in China and Iran, and these were associated with chromosomal abnormalities. The rate of foetal death was not higher in countries like Wales or Lebanon with different health care practices and where invasive foetal testing is not performed when EIF is detected.

## Chromosomal abnormalities

Ten studies [18, 19, 22, 27, 31, 32, 34–37] reported chromosomal abnormalities and the rate of abnormalities was 1.98% (range 0–5.7%) among those with EIF. The chromosomal abnormalities identified with EIF were Trisomy 18, 46, 21 XXY, XYY and Turner's syndrome. Three studies also reported abnormalities among those without EIF and in each of these studies the internal comparison of abnormality rate between EIF and non-EIF suggested no higher rate of chromosomal abnormalities among those with EIF (Fig 4). The sample size of EIF pregnancies were 58 in Ko et al. [22], 1099 in Wang et al. [18] and 230 in Mirza et al. [32].

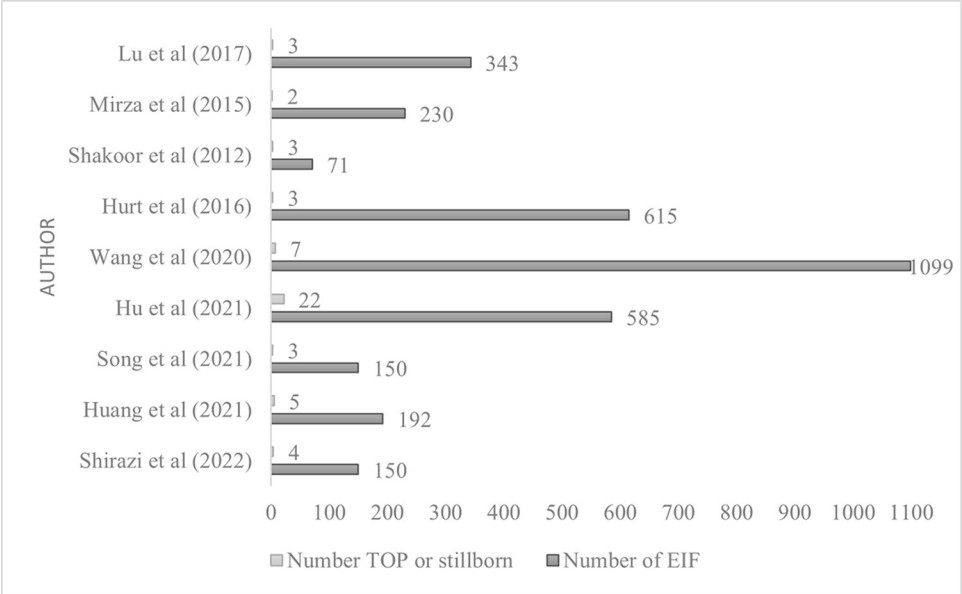

**Fig 3. Studies reporting termination of pregnancy or foetal death [53].**

## Cardiac abnormalities

Cardiac abnormalities were reported in 10 studies [1, 3, 14, 15, 27, 34, 37–39] and the average for all studies was 8.2% (range 1.8% to 39%). When all findings were combined there was a 5.1% rate (224/4385) (Fig 5). The retrospective cohort study findings gave a rate of 3.3% and the prospective studies (with higher rates of missing data and loss to follow-up) was 17.85%. Abnormalities included VSD, tricuspid regurgitation and mitral regurgitation. Two studies reported cardiac abnormalities in non-EIF pregnancies, which ranged from 1.5–3.1%. CHD is thought to affect about <1% of births [40]. There does appear to be evidence that EIF is

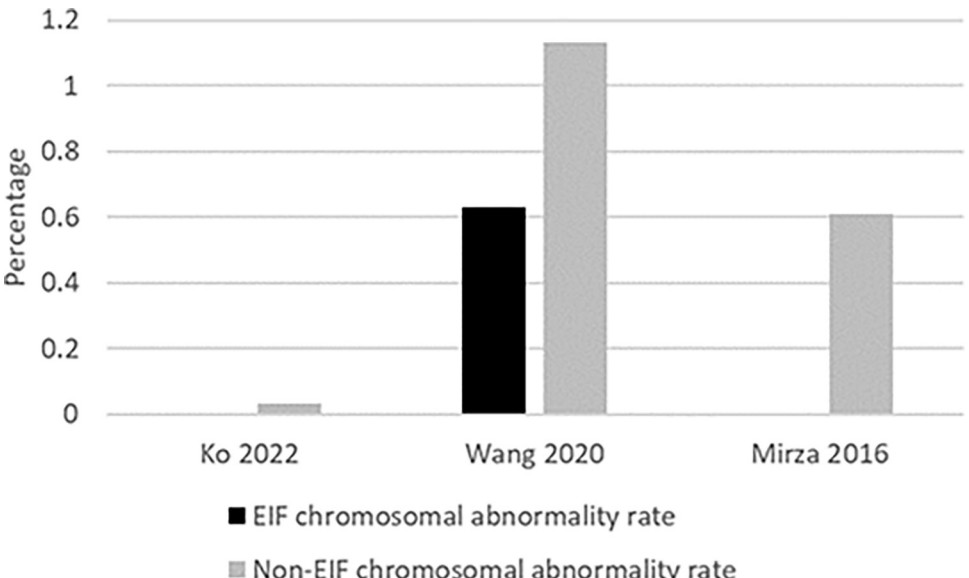

**Fig 4. Comparison of chromosomal abnormality rate among those with EIF and non-EIF.**

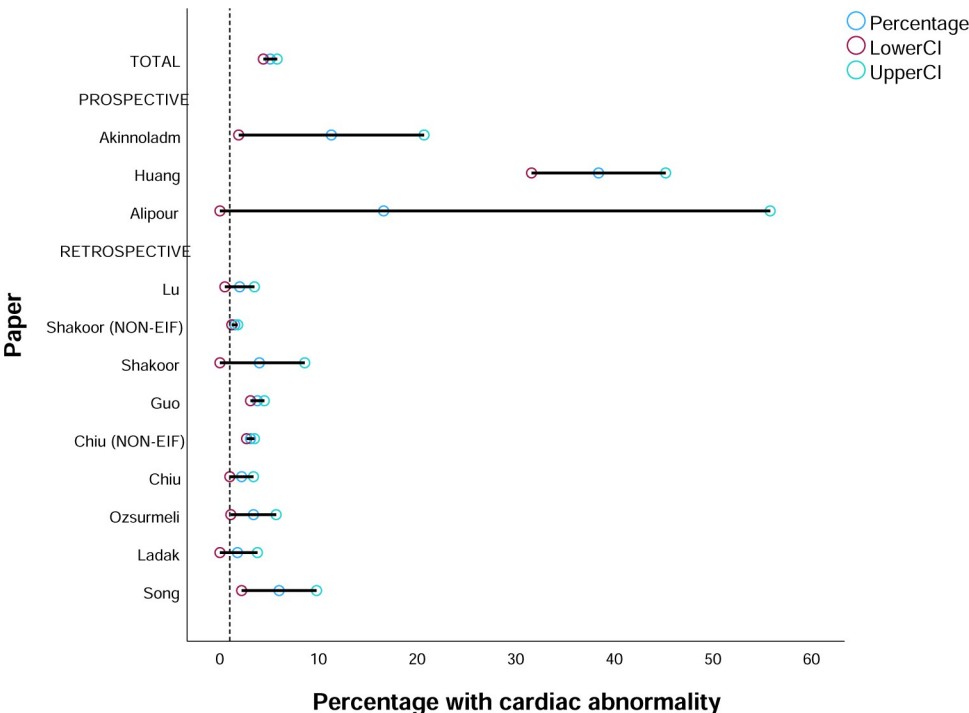

**Fig 5. Proportion of EIF pregnancies with cardiac abnormalities (reference line based on proportion in non-EIF in Shakoor and Chiu)** [1].

associated with higher rates of cardiac abnormality (e.g. 3.3% (95%CI 1.1–5.6) rate of cardiac abnormality). However, it remains that 97% of those detected with EIF will not have a cardiac abnormality. Specific findings from the included studies are available in the supplementary information (S2 Table).

## Location of EIF

The location of the EIF was recorded in 6 studies. However, follow-up of proportion with a cardiac abnormality by EIF location was only recorded in three studies [14, 15, 39]. Combining the findings for left EIF compared to right EIF showed 7.9% (6/76) (95% CI: 2.9%-16.4%) of right EIF were associated with cardiac abnormality compared to 3.6% (107/2,992) (95% CI: 2.9%-4.3%) in the left (difference: 4.3% (95%CI: 0.03% to 12.6%)) showing higher rates with right-sided EIF compared to the left [14, 15, 39]. The rate of abnormality in bilateral EIF did not differ from the rate seen in the left EIF. The findings from Chiu [39] suggested persistent EIF into the third trimester may be associated with higher risk of congenital heart defects such as VSD. Therefore, the follow up of EIF later in pregnancy may be useful to examine if the EIF resolves or persists.

In summary, a EIF in the right ventricle may not resolve as often as that in the left and combining the findings from studies suggested that an EIF in the right is associated with higher rates of cardiac abnormality (7.9% rate if EIF at 20 weeks but this could be higher if persistent beyond 20 weeks). There is ethnic variation in the prevalence of EIF and therefore there is likely to be ethnic variation in the significance of EIF. Only an EIF observed in the third trimester on the right may be associated with cardiac abnormality. However, this would be only 4 per 10,000 pregnancies who would have a right EIF which did not resolve by the third trimester (e.g. 4.5 pregnancies have an EIF, 3% of these are on the right (13.5 per 10,000) and of these

30% persist to third trimester or 4 per 10,000 pregnancies have a persistent EIF on the right). Therefore, EIF in general, is unlikely to be significant as a predictive or screening tool.

## Discussion

Findings from this review highlight that only if the EIF is on the right side and persists until the third trimester, is there a higher chance of cardiac abnormalities (7% risk of cardiac abnormality at 20-week scan), and only in these persistent cases could it be advised that there should be further investigation. This finding that the EIF is not predictive of poor outcomes is significant, especially given the current post COVID-19 strains on health systems worldwide. It must be noted that this data is for isolated markers. There is clear and consistent data that multiple markers are associated with adverse outcomes.

Supportive evidence in terms of the risk of aneuploidy reported that isolated EIF is considered an incidental finding and does not warrant amniocentesis, but that amniocentesis should be undertaken in high-risk cases where other soft markers are identified or there is a history of chromosomal abnormalities in previous pregnancies [38]. A meta-analysis for the risk of Down's Syndrome suggests that EIF increases the risk by 5–7 folds [41]. In contrast, an association between EIF and Trisomy 21 is not always reported [42]. The Society for Foetal Medicine [17] recommends that for those with no previous aneuploid screening and have an isolated EIF, there should be genetic counseling to estimate the probability of Trisomy 21 and discussion on options for non-invasive screening.

Findings from this review suggest that an isolated EIF would not warrant further investigation, that multiple foci do not carry a higher risk compared to single foci, and that recommendations to follow-up should be based on other risk factors (e.g., maternal age, family history) rather than the presence of EIF without other soft markers. It was found that there were higher terminations in countries like China where many pregnant women with EIF have invasive tests to screen for abnormalities. This suggests that EIF in these countries may be randomly associated with or a consequence of chromosomal abnormalities rather than being a marker for it or that simply the perceived risk of future abnormalities is enough to lead to a termination. The majority of infants terminated in these studies had Variants of Uncertain Significance (VOUS) and so may have been healthy. In countries where the presence of EIF does not lead to chromosomal testing there was no evidence of higher rates of death among infants with EIF. In low-risk pregnancies with EIF, invasive chromosomal testing may lead to harm, such as an increased abortion rate, and is unlikely to lead to benefits of increased detection of chromosomal abnormalities.

VSD was associated with EIF in some studies [1, 14, 15, 27, 29, 34, 38, 39] but this association can also be random and non-causative or consequential. VSD is the most common congenital heart defect in children [15, 27, 41, 43, 44]. These often resolve and close spontaneously when the baby is born due to the reversal of vascular resistance at birth. Studies have suggested that overall cardiac function may not be affected by the presence of an isolated EIF when assessed by conventional echocardiography and Tissue Doppler (TD) imaging [45]. Therefore, in isolation, an EIF may not warrant future follow-up. Previous studies have found an association between the presence of an EIF and the incidence of mitral/tricuspid regurgitation. However, it has been suggested that care needs to be taken when assessing if this is true colour flow generated by atrioventricular valvular regurgitation jets or in fact colour Doppler twinkling artefact (which causes a rapidly changing combination of red and blue complexes behind the EIF which may mimic flow) [46]. Transient and mild valvular regurgitation can also be seen in fetuses with no cardiovascular anomaly and can be transient and without pathological significance [47]. Previous research has confirmed rough surfaces, such as that created by

microcalcifications or mineralisation in the papillary muscle (which is what EIF is suggested to be) [48, 49] is seen to reflect the ultrasound incidence beam and increase pulse duration of received radiofrequency. This may be misinterpreted as movement with the atrioventricular valve when visualised on a foetal echocardiogram [46]. Finally, the work by Hurt et al. [31] showed that when the EIF's were taken to a quality assessment panel, 30% were not confirmed as EIF and were removed from the research analysis. Therefore, the identification of EIF has poor inter-rater reliability.

That there is a link between EIF and cardiac abnormality is supported by Taksande et al. [50] who recently published a systematic review and meta-analysis and found a prevalence of 4.8% among studies published between 1987–2021. This is comparable to our finding which included 7 of the same studies (but not a meta-analysis) from 2013–2023. The Taksande [50] analysis did not look at left vs right EIF and did not compare with those without EIF. Our findings suggest, when compared to those without EIF in the same study, only a right sided EIF carries significantly higher risk of cardiac abnormality. A follow-up study by Hurt et al. [51] published 2023, which followed up the infants with EIF and without EIF for 10 years found no evidence of association of EIF with cardiac admissions or diagnosis of cardiac abnormalities but did find an increased risk of respiratory admissions (Hazard ratio 1.27 (95% CI: 1.04–1.54)). The Hurt study did not look at left vs right EIF but the association with respiratory conditions may support the hypothesis that right sided EIF carries morbidity risk more than left sided EIF.

## Strengths and limitations

This work was able to examine outcomes from 15,600 EIFs and reported that EIF at 20 weeks is not significant as a screening tool in pregnancy for infant outcomes at birth. The review covered nine different countries so provides assessment of outcomes for infants of different ethnicities and thus is generalisable. However, the finding that EIF on the right side could carry a 7% risk of cardiac abnormality was based on only two studies and a total of 76 EIFs on the right side. This represents a small sample size which displays high variability in terms of the likelihood of cardiac abnormality (2.9% to 16.9%). An additional study [52], which was excluded as it was considered that the years of recruitment overlapped with Chiu, 2019 [39] and so some pregnancies may be replicated in both papers, found a significantly higher prevalence of congenital heart defect in the right ventricle (14.8%, n = 4 cases) compared to left (2.8%, n = 18 cases) (p = 0.0146). In this paper the cases of congenital heart defect were confirmed by postnatal echocardiography and 16.7% (4/24) chose termination of pregnancy upon detection of structural defect. Absence of prospective and contemporary data in the Western world owing to diminished attention to benign EIF may have caused an unintended bias towards ethnic disparity in the current analysis.

This review was not able to examine the risks of persistent right EIFs, which are likely to affect fewer infants but have a higher likelihood of cardiac abnormality. In addition, the variation in studies from different countries means that combining study outcomes to a single score may be prone to error, in terms of concealing the variation which may exist between different ethnic groups.

The studies included in this review used different health care systems and therefore may not be comparable in their definition and identification of adverse infant outcomes. Combining the findings from different studies may also create errors and biases as there is no harmonisation in their definitions of the outcomes. The studies undertaking prospective analysis were prone to missing data and skewed data collection e.g., only those families who returned for assessment were included. Therefore, the prospective studies were rated as weaker and prone to bias compared to the retrospective studies.

The screening process and review was conducted by two independent reviewers performing the extraction in parallel and then comparing decisions to reduce bias. The review benefited from the use of specialist software Covidence to undertake the review [25]. It allows screening to be more efficient and easily tracked. The date restriction of the last 10 years was justified due to improvements in ultrasound technology in recent years. However, this exclusion criteria may have meant that important previous research was missed.

## Recommendations

Persistent EIF in the right side should be investigated using an echocardiogram. There are limited cases and this could help predict cardiovascular abnormality as there is a higher prevalence in this group. However, there is no justification to follow up EIFs on the left and especially those in isolation with no other markers detected. Any recommendation of follow-up would be based on other risk factors such as multiple other soft markers or family history, rather than the presence of an EIF.

This review did not examine the long-term follow-up of infants with EIF and it is possible that persistent EIF could be associated with adult conditions such as VSD or valve regurgitation. Future investigation into failure to thrive, a possible audible murmur, or symptoms such as breathlessness in an infant, may identify longer term adverse outcomes associated with EIF.

## Conclusions

In conclusion, we examined the outcomes from 19 studies and present that left sided EIFs are not predictive of poor foetal outcomes, and studies reviewed here suggested these are often transient and resolve with time. Persistent EIF on the right side, which is a rare event in itself, would warrant follow-up. This information is beneficial for pregnant women, families, and clinicians to facilitate the development of appropriate clinical guidelines and care pathways during pregnancy and after birth, including the appropriate use of medical interventions and treatment.

## Supporting information

**S1 Table. Extracted variables, definitions and how they were recorded.**
(DOCX)

**S2 Table. Specific findings from included studies which examine cardiac outcomes.**
(DOCX)

**S1 Data.**
(XLSX)

## Author Contributions

**Conceptualization:** Serica Battaglia.

**Data curation:** Serica Battaglia, Lisa Hurt.

**Formal analysis:** Hope Eleri Jones, Serica Battaglia, Lisa Hurt, Sinead Brophy.

**Investigation:** Hope Eleri Jones, Serica Battaglia, Sinead Brophy.

**Methodology:** Serica Battaglia, Sinead Brophy.

**Project administration:** Sinead Brophy.

**Validation:** Hope Eleri Jones.

**Visualization:** Hope Eleri Jones, Serica Battaglia, Sinead Brophy.

**Writing – original draft:** Hope Eleri Jones, Serica Battaglia, Sinead Brophy.

**Writing – review & editing:** Hope Eleri Jones, Serica Battaglia, Lisa Hurt, Orhan Uzun, Sinead Brophy.

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
