## [Decision Letter · Decision Letter 0]

7 Dec 2023

PONE-D-23-34927Echogenic intracardiac foci detection and location in the second-trimester ultrasound and association with fetal outcomes: A systematic literature reviewPLOS ONE

Dear Dr. Jones,

Thank you for submitting your manuscript to PLOS ONE. After careful consideration, we feel that it has merit but does not fully meet PLOS ONE’s publication criteria as it currently stands. Therefore, we invite you to submit a revised version of the manuscript that addresses the points raised during the review process.Almost all issues identified by the reviewers should be addressed in the revision to move forward with this manuscript. Please carefully consider each and every point raised. Please submit your revised manuscript by Jan 21 2024 11:59PM. If you will need more time than this to complete your revisions, please reply to this message or contact the journal office at plosone@plos.org. Please include the following items when submitting your revised manuscript:A rebuttal letter that responds to each point raised by the academic editor and reviewer(s). You should upload this letter as a separate file labeled 'Response to Reviewers'.A marked-up copy of your manuscript that highlights changes made to the original version. You should upload this as a separate file labeled 'Revised Manuscript with Track Changes'.An unmarked version of your revised paper without tracked changes. You should upload this as a separate file labeled 'Manuscript'.If applicable, we recommend that you deposit your laboratory protocols in protocols.io to enhance the reproducibility of your results. Protocols.io assigns your protocol its own identifier (DOI) so that it can be cited independently in the future. For instructions see: https://journals.plos.org/plosone/s/submission-guidelines#loc-laboratory-protocols. Additionally, PLOS ONE offers an option for publishing peer-reviewed Lab Protocol articles, which describe protocols hosted on protocols.io. Read more information on sharing protocols at https://plos.org/protocols?utm_medium=editorial-email&utm_source=authorletters&utm_campaign=protocols.

We look forward to receiving your revised manuscript.

Kind regards,

Mateen A. Khan, Ph.D.

Academic Editor

PLOS ONE

Journal Requirements:

3. Please include a copy of Table 2 which you refer to in your text on page 13.

Reviewers' comments:

Reviewer's Responses to Questions

**Comments to the Author**

1. Is the manuscript technically sound, and do the data support the conclusions?

Reviewer #1: Yes

Reviewer #2: Yes

2. Has the statistical analysis been performed appropriately and rigorously? 

Reviewer #1: Yes

Reviewer #2: N/A

3. Have the authors made all data underlying the findings in their manuscript fully available?

Reviewer #1: Yes

Reviewer #2: Yes

4. Is the manuscript presented in an intelligible fashion and written in standard English?

Reviewer #1: Yes

Reviewer #2: Yes

5. Review Comments to the Author

Reviewer #1: This is an exciting study, as the authors have assembled data from past published papers using the Critical Apparasal Skills Programme methodology. The material is generally well written and structured. This review has tried to establish that echogenic intracardiac foci detection and location alone are not associated with adverse outcomes for infants. I especially think that the inclusion and exclusion criteria were very strict but appropriate.The left-sided EIFs are not predictive of poor fetal outcomes, as this information is beneficial for pregnancy and facilitates appropriate clinical guidelines for appropriate medical intervention.

Reviewer #2: “ Echogenic intracardiac foci detection and location in the second-trimester ultrasound and association with fetal outcomes: A systematic literature review “ well drafted and transparent literature review is to examine the association between EIF and infant outcomes.

Minor points:

To the manuscript, I suggest including the study PMID: 32000445. I would like to suggest that authors compare the yield of chromosomal abnormalities if they consider that it is irrelevant.

1. More fetal echocardiographic characteristics (such as LV, RV, CO, CCO, CCI, SV, FS, etc.) should be included in the beginning, in my opinion.

2. Page 4: Add the title “Introduction “

6. PLOS authors have the option to publish the peer review history of their article (what does this mean?). If published, this will include your full peer review and any attached files.

Reviewer #1: **Yes: **Dr. Abdul Ahad Shaikh

Reviewer #2: **Yes: **Sateesh Maddirevula

---

## [Author Response · Author response to Decision Letter 0]

4 Jan 2024

Amendments have been made to meet the requirements.

3. Please include a copy of Table 2 which you refer to in your text on page 13.

Thank you for this comment, this table is not presented in this manuscript, so this has been removed 

The reference list has been updated and now includes 2 new recently published articles. 

Reviewer #1: This is an exciting study, as the authors have assembled data from past published papers using the Critical Appraisal Skills Programme methodology. The material is generally well written and structured. This review has tried to establish that echogenic intracardiac foci detection and location alone are not associated with adverse outcomes for infants. I especially think that the inclusion and exclusion criteria were very strict but appropriate. The left-sided EIFs are not predictive of poor fetal outcomes, as this information is beneficial for pregnancy and facilitates appropriate clinical guidelines for appropriate medical intervention.

Reviewer #2: “Echogenic intracardiac foci detection and location in the second-trimester ultrasound and association with fetal outcomes: A systematic literature review “well drafted and transparent literature review is to examine the association between EIF and infant outcomes.

Minor points:

To the manuscript, I suggest including the study PMID: 32000445. I would like to suggest that authors compare the yield of chromosomal abnormalities if they consider that it is irrelevant.

We apologise, we had included this reference, but this was not clear as we had not listed the reference to the 10 included studies in the text, we have now added the references to the text to make this clear:

Results section: “Ten studies [22, 35, 28, 19, 36, 37, 38, 18, 32, 39] reported chromosomal abnormalities” 

PMID 32000445 is reference 37 in this list or He et al, 2020. 

1. More fetal echocardiographic characteristics (such as LV, RV, CO, CCO, CCI, SV, FS, etc.) should be included in the beginning, in my opinion.

Thank you very much for this suggestion, we have examined the echocardiographic characteristics of Left ventricle and Right ventricle, multiple and singular. However, there were not enough publications that examined cardiac output (CO), combined cardiac output (CCO), cardiac index (CI), combined cardiac index (CCI), stroke volume (SV) or fractional shortening (FS). They are available in one recent study published after our review has finished - Zhang et al 2023 [1]. This study looked at 59 women with hypothyroidism and found there was cardiac remodeling in fetuses with a mother with hypothyroidism and EIF was detected more often in those with hypothyroidism. However, the article did not report if the remodeling was associated with those with EIF. This study is on a highly selected population (women with hypothyroidism) and so we have not included it in our review, as we can not detect outcome by EIF status. 

1. Yanhong Zhang, Lisi Zhang, Wei Zhao, Na Li, Guihong Chen, Jun Ge, Xingna Su, Shuping Ge & Congxin Sun (2023) Cardiac structural and functional remodeling in the fetuses associated with maternal hypothyroidism during pregnancy, The Journal of Maternal-Fetal & Neonatal Medicine, 36:1, DOI: 10.1080/14767058.2023.2203796

We have now updated our discussion to include recent findings from papers published after the dates of our review. The following has been added to the discussion: 

That there is a link between EIF and cardiac abnormality is supported by Taksande et al [54] who recently published a systematic review and meta-analysis and found a prevalence of 4.8% among studies published between 1987-2021. This is comparable to our finding which included 7 of the same studies (but not a meta-analysis) from 2013-2023. The Taksande [54] analysis did not look at left vs right EIF and did not compare with those without EIF. Our findings suggest, when compared to those without EIF in the same study, only a right sided EIF carries significantly higher risk of cardiac abnormality. A follow-up study by Hurt et al [55] published 2023, which followed up the infants with EIF and without EIF for 10 years found no evidence of association of EIF with cardiac admissions or diagnosis of cardiac abnormalities but did find an increased risk of respiratory admissions (Hazard ratio 1.27 (95% CI: 1.04-1.54). The Hurt study did not look at left vs right EIF but the association with respiratory conditions may support the hypothesis that right sided EIF carries morbidity risk more than left sided EIF. 

2. Page 4: Add the title “Introduction “

This has been added

---

## [Decision Letter · Decision Letter 1]

24 Jan 2024

Echogenic intracardiac foci detection and location in the second-trimester ultrasound and association with fetal outcomes: A systematic literature review

PONE-D-23-34927R1

Dear Dr. Jones,

We’re pleased to inform you that your manuscript has been judged scientifically suitable for publication and will be formally accepted for publication once it meets all outstanding technical requirements.

Kind regards,

Mateen A. Khan, Ph.D.

Academic Editor

PLOS ONE

Additional Editor Comments (optional):

Reviewers' comments:

Reviewer's Responses to Questions

**Comments to the Author**

1. If the authors have adequately addressed your comments raised in a previous round of review and you feel that this manuscript is now acceptable for publication, you may indicate that here to bypass the “Comments to the Author” section, enter your conflict of interest statement in the “Confidential to Editor” section, and submit your "Accept" recommendation.

Reviewer #2: All comments have been addressed

2. Is the manuscript technically sound, and do the data support the conclusions?

Reviewer #2: Yes

3. Has the statistical analysis been performed appropriately and rigorously? 

Reviewer #2: N/A

4. Have the authors made all data underlying the findings in their manuscript fully available?

Reviewer #2: Yes

5. Is the manuscript presented in an intelligible fashion and written in standard English?

Reviewer #2: Yes

6. Review Comments to the Author

Reviewer #2: Authors have addressed all my concerns. Authors have addressed all my concerns. Authors have addressed all my concerns.

7. PLOS authors have the option to publish the peer review history of their article (what does this mean?). If published, this will include your full peer review and any attached files.

Reviewer #2: **Yes: **Sateesh Maddirevula

---

## [Editor Report · Acceptance letter]

12 Feb 2024

PONE-D-23-34927R1 

PLOS ONE

Dear Dr. Jones, 

I'm pleased to inform you that your manuscript has been deemed suitable for publication in PLOS ONE. Congratulations! Your manuscript is now being handed over to our production team.

Kind regards, 

on behalf of

Dr. Mateen A. Khan 

Academic Editor

PLOS ONE